# Component of nicotine-induced intracellular calcium elevation mediated through α3- and α5-containing nicotinic acetylcholine receptors are regulated by cyclic AMP in SH-SY 5Y cells

Tamayo Takahashi[1,2], Takayuki Yoshida[3], Kana Harada[1], Tatsuhiko Miyagi[1], Kouichi Hashimoto[3], Izumi Hide[1], Shigeru Tanaka[1], Masahiro Irifune[2], Norio Sakai[1]*

1 Department of Molecular and Pharmacological Neuroscience, Graduate School of Biomedical and Health Sciences, Hiroshima University, Hiroshima, Japan, 2 Department of Dental Anesthesiology, Graduate School of Biomedical and Health Sciences, Hiroshima University, Hiroshima, Japan, 3 Department of Neurophysiology, Graduate School of Biomedical and Health Sciences, Hiroshima University, Hiroshima, Japan

* nsakai@hiroshima-u.ac.jp

**Data Availability Statement:** All relevant data are within the manuscript and its Supporting information files.

## Abstract

The pathway from the medial habenular nucleus to the interpeduncular nucleus, in which nicotinic acetylcholine receptor (nAChR) including the α3 and α5 subunits (α3 * and α5 * nAChRs) are expressed, is implicated in nicotine dependence. We investigated whether α3 * and α5 * nAChRs are regulated by cAMP using SH-SY5Y cells to clarify the significance of these receptors in nicotine dependence. We analyzed the nicotine-induced elevation of intracellular $Ca^{2+}$ ($[Ca^{2+}]i$). Nicotine induces a concentration-dependent increase in $[Ca^{2+}]i$. The elimination of $Ca^{2+}$ from extracellular fluid or intracellular stores demonstrated that the nicotine-induced $[Ca^{2+}]i$ elevation was due to extracellular influx and intracellular mobilization. The effects of tubocurarine on nicotine-induced $[Ca^{2+}]i$ elevation and current suggest that intracellular mobilization is caused by plasma membrane-permeating nicotine. The inhibition of α3 *, α5 *, α7 nAChR and voltage-gated $Ca^{2+}$ channels by using siRNAs and selective antagonists revealed the involvement of these nAChR subunits and channels in nicotine-induced $[Ca^{2+}]i$ elevation. To distinguish and characterize the α3 * and α5 * nAChR-mediated $Ca^{2+}$ influx, we measured the $[Ca^{2+}]i$ elevation induced by nonmembrane-permeating acetylcholine when muscarinic receptors, α7nAChR and $Ca^{2+}$ channels were blocked. Under this condition, the $[Ca^{2+}]i$ elevation was significantly inhibited with a 48-h treatment of dibutyryl cAMP, which was accompanied by the downregulation of α3 and β4 mRNA. These findings suggest that α3 * and α5 * nAChR-mediated $Ca^{2+}$ influx is possibly regulated by cAMP at the transcriptional level.

## Introduction

Nicotinic acetylcholine receptors (nAChRs) are pentameric ion channel-containing receptors composed of α, β, γ, and δ subunits. In the central nervous system (CNS), they are composed

**Funding:** N.S. was supported by JSPS KAKENHI Grant Numbers 19H03409, 16H05133, 16K15318 and 25293061. I. H. was supported by JSPS KAKENHI Grant Numbers 18K06891 and 15K08234. S. T. was supported by JSPS KAKENHI Grant Numbers 18K07392 and 15K09317.

**Competing interests:** The authors have declared that no competing interests exist.

of α2–4 and β2–10 subunits and are homopentamers composed of a single subunit or hetero-pentamers composed of different subunits [1]. Binding of ligands such as acetylcholine and nicotine to nAChR causes the influx of cations such as $Na^+$ and $Ca^{2+}$. The influx of $Ca^{2+}$ triggers a variety of intracellular signaling events and the release of neurotransmitters from nerve terminals [1, 2]. Nicotinic receptors in the CNS have been implicated in the regulation of pain [3–5] and in the formation of nicotine dependence [6, 7].

The dopaminergic nervous system between the ventral tegmental area and the nucleus accumbens is known as a neural pathway that forms drug dependence, including nicotine dependence [6]. When nicotine enters the brain, it binds to nAChRs on dopamine neurons in the ventral tegmental area and is thought to activate the brain reward system by releasing dopamine [8]. The medial habenular nucleus projects to the interpeduncular nucleus and is thought to indirectly control dopamine neurons in the ventral tegmental area [9]. It has been reported that nicotinic receptors containing the α5 subunit (α5 * nAChRs), which are expressed in this neuronal pathway [10], regulate the aversion response to high concentrations of nicotine, suggesting that α5 * nAChR is a target of nicotine dependence [11]. In addition to α5 * nAChRs, α3 * nAChRs are widely expressed in the habenular nucleus [12], implicating these nAChRs in nicotine dependence; however, the regulatory mechanism of these receptors has not been fully elucidated because of the lack of selective agonists and antagonists for universal use.

GPR3, a G protein-coupled receptor that is constitutively associated with Gs (stimulatory G protein) and maintains intracellular cAMP levels, is abundantly expressed in the CNS [13]. In the mouse brain, GPR3 is expressed predominantly in the medial habenular nucleus and hippocampus [14]. GPR3 has been reported to be an important factor in neurite outgrowth, survival, differentiation, and polarization [13, 15, 16].

Although the physiological significance of GPR3 expression in the medial habenular nucleus has not yet been elucidated, it has been reported that cAMP-PKA signaling regulates signal transduction through α3 * nAChRs [17, 18]. This signaling leads to the possibility that α3 * nAChR may be regulated by cAMP, the level of which is increased via GPR3, and this mechanism may be involved in nicotine dependence. In this study, we focused on nicotine-induced intracellular $Ca^{2+}$ ($[Ca^{2+}]i$) elevation because both α3 * and α5 * nAChRs have calcium ion permeability [19, 20]. To elucidate whether α3 * and α5 * nAChRs are regulated by cAMP, we sought to clarify the basic properties of α3 * and α5 * nAChR-mediated $[Ca^{2+}]i$ elevation and the cAMP-mediated regulation of these nAChRs using SHSY-5Y cells, in which both α3 * and α5 * nAChRs were endogenously expressed [14, 21].

## Materials and methods

### Materials

Nicotine was purchased from Nacalai Tesque (Kyoto, Japan). Thapsigargin and ionomycin were purchased from FUJIFILM Wako Pure Chemical Industries, Ltd. Fluo-4-AM and Power-Load™ concentrate were purchased from Molecular Probe (Eugene, OR, USA). Acetylcholine, d-tubocurarine, nifedipine and dibutyryl cyclic AMP were purchased from Sigma-Aldrich (St. Louis, MO, USA). Omega-conotoxin GVIA and α-conotoxin AuIB were from the Peptide Institute, Inc. (Osaka, Japan). Atropine was obtained from Katayma Chemical Industries Co., Ltd. (Osaka, Japan). All the other chemicals used were of analytical grade.

### Cell culture and transfection

SH-SY5Y cells were purchased from the Riken Cell Bank (Tsukuba, Japan) and were maintained in Dulbecco's modified Eagle's medium/Ham's F-12 medium (FUJIFILM Wako,

Osaka, Japan) containing 10% fetal bovine serum, penicillin (100 units/ml), and streptomycin (100 μg/ml). To obtain fluorescence images, cultured cells were seeded onto glass-bottom culture dishes (MatTek Corporation, Ashland, OR, USA).

## RT-PCR

Total RNA was extracted from subconfluent SH-SY 5Y cells using an RNeasy Mini kit (Qiagen, Hilden, Germany) and converted into first-strand cDNA using the QuantiTect reverse transcription kit (Qiagen). Real-time PCR was performed using SYBR Green Real-time PCR Master Mix and an ABI Prism model 7500 sequence detection system (Applied Biosystems, Foster City CA, USA). mRNA level was normalized to that of GAPDH mRNA. The relative expression was calculated using the ΔCT method. The primer set used to detect human α3nAChR subunit was `5'-GCTTGTAGTCATTCCAGATTTGC-3'` and `5'-ACCCAGTCATCATC CATTTCG-3'`. The primer set used to detect human α5nAChR subunit was `5'-CATC TATCCATTCCTGTTTCAACC-3'` and `5'-CCTGTGGAACACCTGAATGAC-3'`. The primer set used to detect human β4nAChR subunit was `5'-GTCCATTCCTGTTTCAGCCA-3'` and `5'-CTCACAGCTCATCTCCATCAAG-3'`.

   The primer set used to detect human GAPDH mRNA, an internal control, was `5'- TGTAGTTGAGGTCAATGAAGGG-3'` and `5'-ACATCGCTCAGACACCATG-3'`. R-squared values of the PCR were 0.9997, 0.9999, 0.9996, 1, 0.9998, 1 for PCRs of α3, α5, α7, β2, β4 and GAPDH, respectively. The amplification efficiencies of the primers used to detect nAChRs mRNA in this study were almost 100% without exception.

## Loading of Fluo-4 and observed increases in intracellular calcium

The culture medium of the cells was replaced with normal HEPES buffer composed of 165 mM NaCl, 5 mM KCl, 1 mM $MgCl_2$, 1 mM $CaCl_2$, 5 mM HEPES, and 10 mM glucose at pH 7.4. To eliminate the extracellular $Ca^{2+}$, 1 mM $CaCl_2$ was replaced with 2.5 mM EGTA. Then, the calcium indicator Fluo-4 (2.5 μM) and PowerLoad™ concentrate (diluted to 1:100) were loaded into cells 30 min prior to observation. To observe nicotine- or acetylcholine-triggered calcium elevation, time-lapse imaging was carried out using a fluorescence microscope (Keyence, BZ-9000) every 15 seconds for 15 min after the application of nicotine or acetylcholine. To observe the effects of drugs on $Ca^{2+}$ elevation, cells were pretreated for 15 min before nicotine or acetylcholine was applied.

## Semiquantitative evaluation of nicotine-induced increases in intracellular calcium

We selected between 8 and 10 regions of interest (ROIs) and determined the average fluorescence change in these regions. The time-dependent change in intracellular calcium ion $[Ca^{2+}]i$ was determined by setting the fluorescence intensity to 100% before administration of nicotine or acetylcholine (Fig 1B). At the end of each trial, ionomycin was administered at a final concentration of 0.5 μM to confirm the loading state of Fluo-4 AM. The increase in $[Ca^{2+}]i$ in each study was quantified as a ratio of the peak increase in $[Ca^{2+}]i$ induced by nicotine to that induced by ionomycin, according to the modified methods in a previous report [2].

## Transfection of siRNA and an expression plasmid for delivering GPR3 into SHSY-5Y cells

For transfection of siRNAs for nAChR and GPR3, Lipofectamine RNAiMAX (Thermo Fisher, Waltham, MI, USA) was used according to the manufacturer's recommended protocol. An

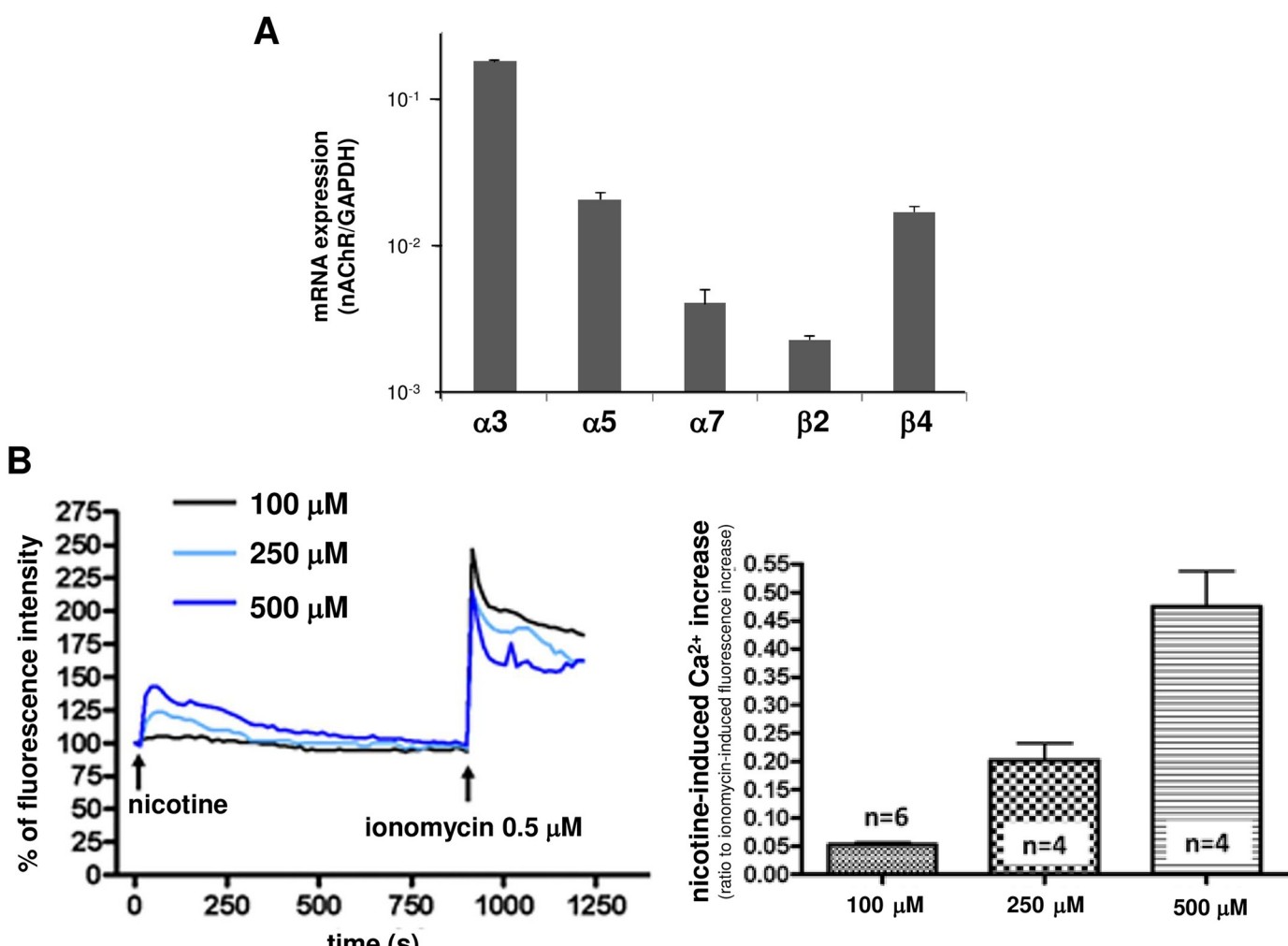

**Fig 1.** (A) Expression of the mRNAs of the nicotinic receptor subunits in SH-SY5Y cells. The type of nicotinic receptor expressed in SH-SY5Y cells was examined by real-time PCR. The expression of mRNA for the α3, α5, α7, β2, and β4 nicotinic receptor subunits were confirmed. Data represent the mean ± standard error of 3 independent experiments. (B) Nicotine-induced intracellular $Ca^{2+}$ ($[Ca^{2+}]i$) elevation in SH-SY5Y cells. Left panel: The $[Ca^{2+}]i$ increase induced by nicotine was observed for 15 min, and ionomycin was administered at the end of observation. The fluorescence before nicotine administration was assumed to be 100%, and the change in $[Ca^{2+}]i$ after application of nicotine at each concentration is shown. Data represent the average of the experiments shown in right panel. Right panel: The ratio of peak $[Ca^{2+}]i$ elevation induced by nicotine to the $[Ca^{2+}]i$ elevation induced by ionomycin was evaluated as an index of nicotine-induced $[Ca^{2+}]i$ elevation. A concentration-dependent nicotine-induced $[Ca^{2+}]i$ increase was observed. The data represent the means ± standard error.

expression plasmid for DsRed was also cotransfected to detect the siRNA-transfected cells. Double-stranded siRNAs corresponding to human GPR3 were purchased from Life Technologies Japan (Tokyo, Japan). The target siRNA sequence was 5′-CUACUAUUCAGAGACAA CAtt-3. The RNA sequence for the control siRNA was 5′-UACUAUUCGACACGCGAA GdTdT-3′. Small interfering RNAs targeting human α3, α5, α7, β2 and β4 nAChR subunits were purchased from Applied Biosystems (Waltham, MA, USA).

## Measurement of nicotine-induced current

Whole-cell recordings were made from the SH-SY5Y cells, and ionic currents were recorded in the voltage-clamp mode at -70 or -60 mV with an Axopatch 200B (Molecular Device, Sunnyvale, CA, USA). The signals were filtered at 3 kHz and digitized at 20 kHz. On-line data

acquisition and off-line data analysis were performed using pClamp and Clampex software (v10.7, Molecular Device), respectively. Nicotine and tubocurarine were bath-applied, and leak currents were sampled every five seconds. The intracellular solution consisted of CsCl, 5 mM; Cs D-gluconate, 125 mM; NaCl, 10 mM; HEPES, 10 mM; EGTA, 10 mM; Mg-ATP, 4 mM; and (2 Na) -GTP, 0.4 mM. The external solution consisted of 125 mM NaCl, 2.5 mM KCl, 2 mM $CaCl_2$, 1 mM $MgSO_4$, 1.25 mM $NaH_2PO_4$, 26 mM $NaHCO_3$, and 20 mM glucose.

## Statistical analysis

Statistics were determined using Prism 4 software (GraphPad Software, San Diego, CA). Statistical significance was determined by unpaired t-test or one-way ANOVA followed by Dunnett's posttest. If the $p$ value was less than 0.05 ($p < 0.05$), the difference was considered significant.

## Results

### Expression of various nicotinic acetylcholine receptor (nAChR) mRNAs in SH-SY5Y cells

Previous reports have shown that SH-SY5Y cells endogenously express the mRNAs for the α3, α5, α7, β2, and β4 nAChR subunits [22]. As shown in Fig 1A, our present study confirmed the presence of these mRNAs as previously reported. The mRNA levels for the α3, α5 and β4 nAChR subunits were more prominent than those for the α7 or β2 nAChR subunits.

### Fundamental properties of nicotine-induced intracellular $Ca^{2+}$ elevation in SH-SY5Y cells

**Nicotine-induced elevation of intracellular $Ca^{2+}$ in SH-SY5Y cells.** We investigated the $[Ca^{2+}]i$ elevation in SH-SY5Y cells after nicotine administration. As shown in Fig 1B, a concentration-dependent increase in $[Ca^{2+}]i$ was observed when nicotine at concentrations between 100 μM and 500 μM was administered.

**Investigation of the origin of $Ca^{2+}$ in nicotine-induced $[Ca^{2+}]i$ elevation.** We examined the origin of $Ca^{2+}$ involved in the nicotine-induced intracellular $[Ca^{2+}]i$ elevation observed in SH-SY5Y cells. First, we examined whether nicotine (500 μM)-induced intracellular $[Ca^{2+}]i$ elevation occurs when extracellular $Ca^{2+}$ was eliminated. As indicated by the light blue line in Fig 2A, the elimination of extracellular $Ca^{2+}$ did not remarkably affect the nicotine-induced intracellular $[Ca^{2+}]i$ elevation, suggesting that nicotine-induced intracellular $[Ca^{2+}]i$ elevation may also be critical for intracellular $Ca^{2+}$ mobilization.

To confirm this finding, we used thapsigargin (5 μM), an endoplasmic reticulum calcium pump inhibitor. As shown in the purple line in Fig 2B, increases in intracellular $[Ca^{2+}]i$ levels were observed when thapsigargin was administered in the presence of extracellular $Ca^{2+}$. In this state, when intracellular $Ca^{2+}$ stores were depleted, nicotine (500 μM) induced an increase in $[Ca^{2+}]i$, which is thought to be the result of extracellular $Ca^{2+}$ influx. The black line indicates that a nicotine-induced $[Ca^{2+}]i$ elevation was also observed when the buffer was administered alone as a vehicle. Then, as indicated by the purple line in Fig 2C, $Ca^{2+}$-free buffer was used to eliminate the extracellular $Ca^{2+}$, and thapsigargin (5 μM) was administered to deplete the intracellular $Ca^{2+}$ stores. Depletion of intracellular and extracellular $Ca^{2+}$ completely abolished the nicotine (500 μM)-induced $[Ca^{2+}]i$ increase (Fig 2C, purple line). When a $Ca^{2+}$-free buffer was administered alone as a vehicle, nicotine-induced $[Ca^{2+}]i$ elevation was observed, which may have been the result of intracellular $Ca^{2+}$ mobilization (Fig 2C, black line). Taken

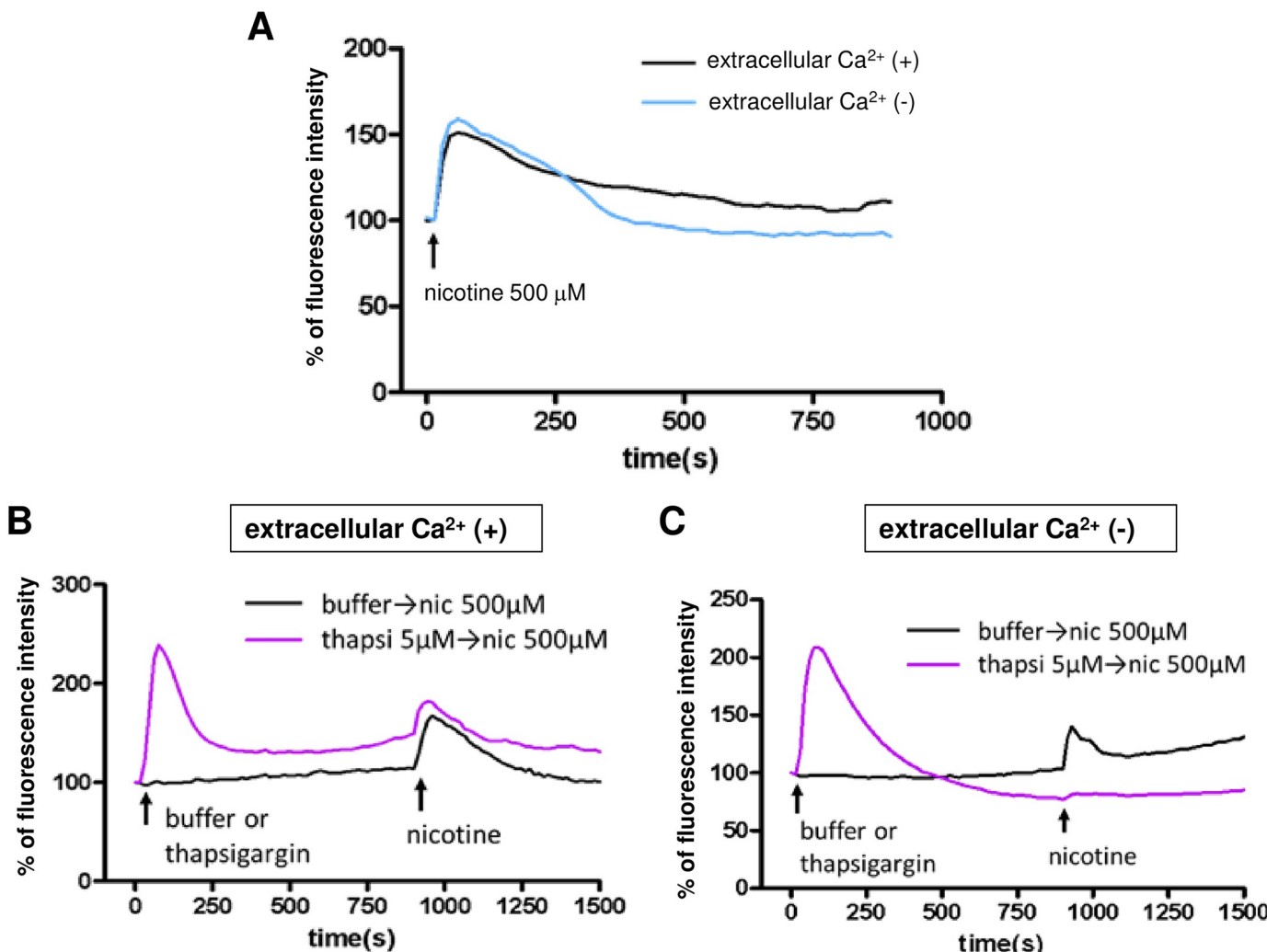

**Fig 2. The origin of the Ca²⁺ involved in the nicotine-induced [Ca²⁺]i elevation in SH-SY5Y cells.** (A) Effect of eliminating extracellular Ca²⁺. A Ca²⁺-free buffer in which calcium was replaced by EGTA was used. The [Ca²⁺]i elevation after 500 μM nicotine treatment was observed when extracellular Ca²⁺ was eliminated. (B) In the presence of extracellular Ca²⁺, 5 μM thapsigargin was incubated for 15 min to deplete the intracellular Ca²⁺ stores. Then, 500 μM nicotine was applied. The [Ca²⁺]i elevation, which is thought to be the result of extracellular influx, was observed (purple line). [Ca²⁺]i elevation was also observed when nicotine was administered without thapsigargin (black line). (C) In the presence of the Ca²⁺ -free buffer, thapsigargin was incubated for 15 min to eliminate intracellular and extracellular Ca²⁺. The subsequent application of nicotine did not induce [Ca²⁺]i increases (purple line). The [Ca²⁺]i elevation was observed when nicotine was administered without thapsigargin, which is thought to mobilize Ca²⁺ from intracellular organelles (black line). A representative example of more than 5 cases is shown in Fig 2A–2C.

together, these results suggest that there are two processes of nicotine-induced [Ca²⁺]i elevation: extracellular influx and intracellular mobilization.

**Effects of nonselective nicotinic receptor antagonists on nicotine-induced [Ca²⁺]i elevation and nicotine-induced current.** To elucidate whether nicotine-induced intracellular [Ca²⁺]i elevation was mediated by nAChRs, we examined the effects of the nonselective nAChR antagonist tubocurarine. We first examined effects of tubocurarine on nicotine-induced currents using whole-cell recording. As shown in Fig 3A, inward currents induced by bath application of 500 μM nicotine were almost completely inhibited by pretreatment with 100 μM tubocurarine. Tubocurarine (100 μM) also completely inhibited the nicotine

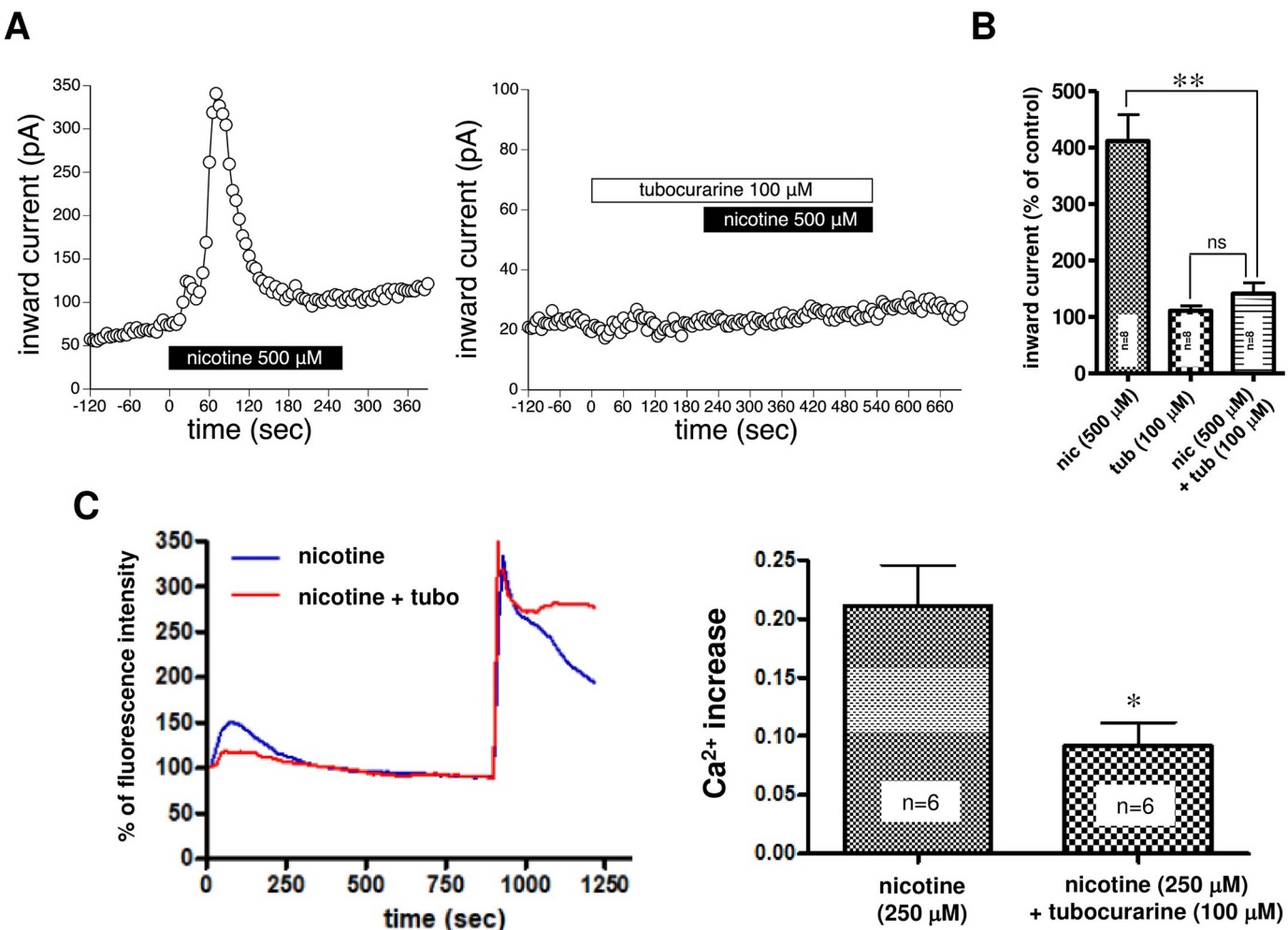

**Fig 3. Effects of tubocurarine, a nonselective nAChR antagonist, on nicotine-induced intracellular [Ca$^{2+}$]i elevation and current.** (A) The representative time plot of leak currents at -70 or -60 mV. Leak currents were sampled every ten seconds. Left panel: The bath application of nicotine (500 μM) evoked the large inward current in a SH-SY5Y cell. Nicotine was applied during a period indicated by a black line. Right panel: Pretreatment with 100 μM tubocurarine (white line) completely suppressed the nicotine-induced current. Representative examples from 8 independent experiments are presented. (B) Quantitative analysis of the effects of tubocurarine on nicotine (500 μM)-induced currents. The average of three consecutive data just prior to drug administration was set at 100% and the inward current under each condition was normalized. Abbreviations, nic and tub, indicate nicotine and tubocurarine, respectively. $^{**}$p<0.01, ns p>0.05, one-way ANOVA followed by Dunnett's posttest, n = 8. (C) Left panel: We investigated the temporal changes in nicotine (250 μM)-induced [Ca$^{2+}$]i elevation following 15 min of pretreatment with the nonselective nAChR antagonist tubocurarine (100 μM). Black line: 250 μM nicotine alone, Blue line: 15 min pretreatment with 100 μM tubocurarine. Data represent the average of the six experiments shown in right panel. Right panel: Ca$^{2+}$ increases in the 250 μM nicotine alone-treated group and the 15 min pretreatment with 100 μM tubocurarine groups are shown. Data indicate the means ± standard error. $^{*}$ p < 0.05, unpaired t-test vs the nicotine-alone-treated group.

(250 μM)-induced current (S1 Fig). In quantitative analysis, 100 μM tubocurarine significantly and robustly inhibited nicotine (500 μM)-induced currents (Fig 3B). The nicotine (500 μM)-induced currents in the presence of tubocurarine were not significantly different from currents in the presence of tubocurarine alone (Fig 3B). These results suggest that the nicotine-induced current is largely mediated by nAChRs.

Next we examined effects of tubocurarine on the nicotine (250 μM)-induced [Ca$^{2+}$]i elevation. As shown in Fig 3C, tubocurarine significantly suppressed nicotine-induced [Ca$^{2+}$]i elevation, which suggests that [Ca$^{2+}$]i elevation is largely mediated by nAChRs expressed in plasma membrane. Unexpectedly, the small, but apparent [Ca$^{2+}$]i elevation was still remained

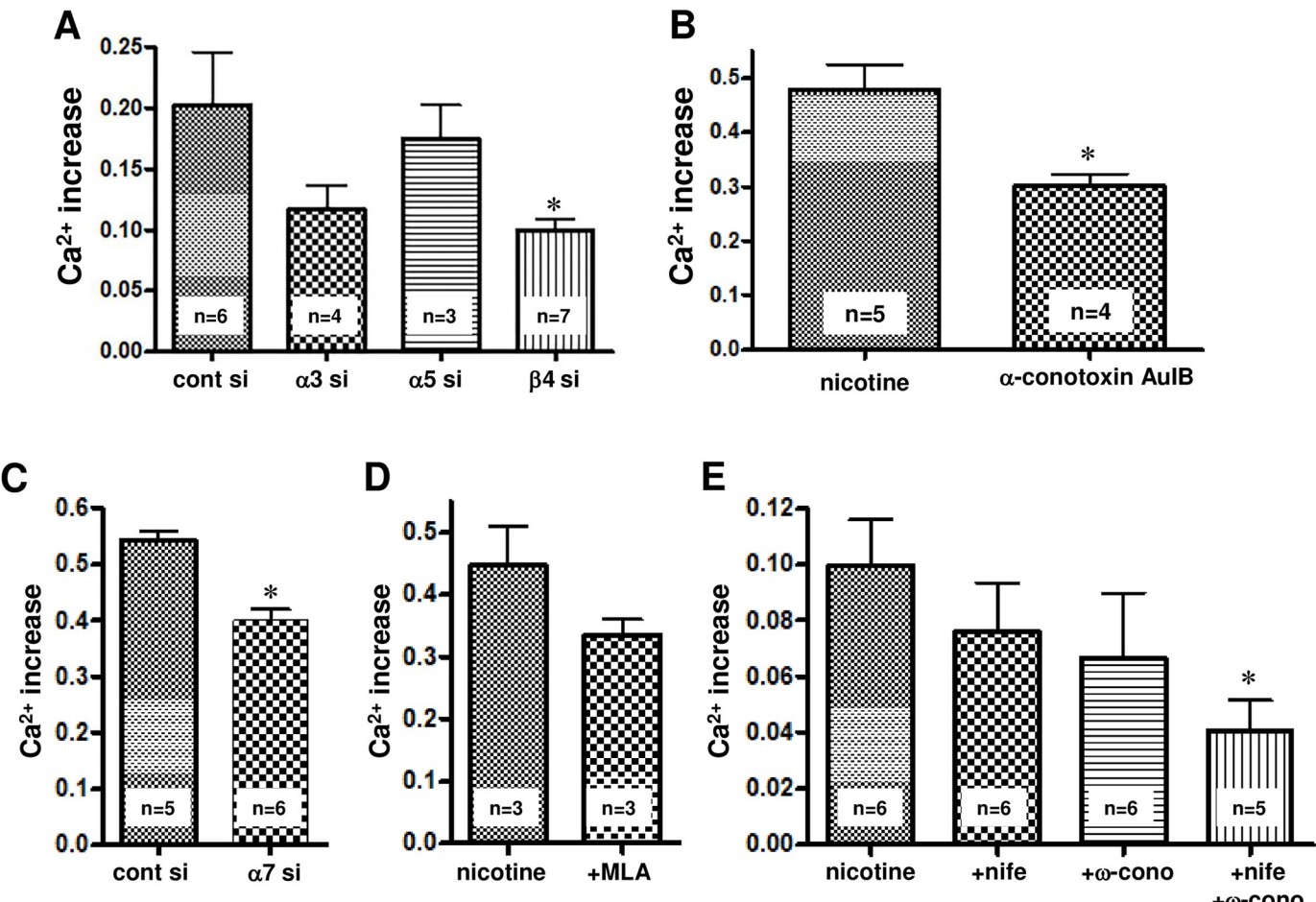

**Fig 4. Involvement of various nAChR subtypes and voltage-gated calcium channels in nicotine (250 μM)-induced [Ca²⁺]i elevation.** (A) Effects of siRNA treatment of nAChR subtypes (α3, α5 and β4) on nicotine-induced [Ca²⁺]i elevation. (B) Effect of the α3β4nAChR-selective antagonist α-conotoxin AuIB on nicotine-induced [Ca²⁺]i elevation. (C) The effect of siRNA treatment of the α7nAChR subunit on nicotine-induced [Ca²⁺]i elevation. (D) The effect of MLA (5 μM), a selective α7nAChR antagonist, on nicotine-induced [Ca²⁺]i elevation. Cells were pretreated with MLA for 15 min before 250 μM nicotine was added. (E) The effects of 5 μM nifedipine, an L-type Ca²⁺ channel antagonist, and 1 μM ω-conotoxin GVIA, an N-type Ca²⁺ channel antagonist, on nicotine (250 μM)-induced [Ca²⁺]i elevation. The cells were pretreated with inhibitors for 15 min before 250 μM nicotine was added. The data represent the means ± standard error. * p< 0.05, unpaired t-test, vs the nontreated control group or the control siRNA-treated group.

in the presence of tubocurarine (Fig 3C). Because nicotine is known to be membrane–permeable [23], penetrated nicotine might induce intracellular calcium mobilization by unknown mechanism. These findings also support the idea that a component in the intracellular calcium mobilization process contributes to nicotine-induced [Ca²⁺]i elevation.

**Involvement of nAChRs in nicotine-induced [Ca²⁺]i elevation.** As shown in Fig 1, the results indicate that SH-SY5Y cells expressed α3, α5, α7, β2 and β4 nAChR subunits. Therefore, the involvement of each subunit in nicotine-induced calcium elevation was elucidated using siRNA for each subunit or specific nAChR inhibitors. As shown in Fig 4A, the siRNA for the β4 nAChR subunit significantly reduced the nicotine (250 μM)-induced [Ca²⁺]i elevation compared with the control in siRNA-transfected cells. The small interference RNA for the α3 nAChR subunit tended to decrease the nicotine-induced [Ca²⁺]i elevation, but not significantly. Since β4 nAChR functions as a core subunit of α3 * and α5 * nAChRs, these results suggest the involvement of β4-containing nAChRs, such as α3β4 and α5β4 nAChR, in nicotine-

induced [$Ca^{2+}$]i elevation. This idea is supported by the finding showing that α-conotoxin AuIB (10 μM), an α3β4 nAChR-specific inhibitor, significantly inhibited the increase in nicotine (500μM)-induced [$Ca^{2+}$]i level (Fig 4B).

In addition, as shown in Fig 4C, the siRNA for the α7 nAChR subunit significantly decreased the nicotine (250 μM)-induced [$Ca^{2+}$]i elevation. Methyllycaconitine (MLA, 5 μM), a specific α7 nAChR inhibitor, tended to inhibit nicotine (250 μM)-induced [$Ca^{2+}$]i elevation but not significantly (Fig 4D). These results suggest the involvement of the α7 nAChR in nicotine-induced [$Ca^{2+}$]i elevation in SH-SY 5Y cells.

**Involvement of voltage-gated $Ca^{2+}$ channels in nicotine-induced [$Ca^{2+}$]i elevation.** SH-SY5Y cells are known to express L-type and N-type voltage-gated $Ca^{2+}$ channels [21]. We investigated the effects of 5 μM nifedipine, an L-type $Ca^{2+}$ channel antagonist, and 1 μM ω-conotoxin GVIA, an N-type $Ca^{2+}$ channel antagonist, on nicotine (250 μM)-induced [$Ca^{2+}$]i elevation. As shown in Fig 4E, although a single dose of nifedipine or ω-conotoxin GVIA did not significantly reduce the nicotine-induced [$Ca^{2+}$]i elevation, simultaneous administration of both inhibitors significantly inhibited the nicotine-induced [$Ca^{2+}$]i elevation (Fig 4E). This suggests that nicotine-induced [$Ca^{2+}$]i elevation involves voltage-gated $Ca^{2+}$ channels.

## Properties of acetylcholine-induced [$Ca^{2+}$]i elevation in SH-SY5Y cells

In contrast to nicotine which is membrane permeable, acetylcholine, which does not permeate the plasma membrane, suggests that the selective induction of [$Ca^{2+}$]i elevation is mediated through nAChRs expressed in the plasma membrane. However, because acetylcholine acts not only on nAChRs but also on muscarinic receptors, the use of atropine, a muscarinic receptor antagonist, is necessary to observe the acetylcholine-induced response mediated solely through nAChRs.

**Effects of atropine on acetylcholine-induced [$Ca^{2+}$]i elevation in SH-SY5Y cells.** We first examined the nature of the acetylcholine-induced [$Ca^{2+}$]i elevation. As shown in Fig 5A, the [$Ca^{2+}$]i increase induced by 10 μM acetylcholine alone was more pronounced than the [$Ca^{2+}$]i increase induced by nicotine, showing a steep rise and a slow fall. Because acetylcholine acts on muscarinic receptors and nAChR, the increase in [$Ca^{2+}$]i is thought to include a muscarinic receptor-mediated response. On the other hand, as shown in Fig 5B, when 10 μM acetylcholine was coadministered with 1 μM atropine to stimulate only nAChRs, the [$Ca^{2+}$]i elevation immediately decreased after a steep rise, which differed from the pattern of acetylcholine alone-induced [$Ca^{2+}$]i elevation. This sharp increase and decrease in [$Ca^{2+}$]i happened faster than the nicotine-induced [$Ca^{2+}$]i elevation.

**Involvement of voltage-gated $Ca^{2+}$ channels and α7nAChR in acetylcholine-induced [$Ca^{2+}$]i elevation in the presence of atropine.** We next examined whether voltage-gated $Ca^{2+}$ channels and α7nAChRare involved in acetylcholine (10 μM)-induced [$Ca^{2+}$]i elevation in the presence of atropine (1 μM). As shown in Fig 5C, pretreatment with 5 μM nifedipine and 1 μM ω-conotoxin GVIA for 15 min significantly inhibited the increase in [$Ca^{2+}$]i. As shown in Fig 5D, pretreatment with the α7nAChR-specific antagonist MLA (5 μM) significantly inhibited the [$Ca^{2+}$]i increase. These results suggest that the voltage-gated $Ca^{2+}$ channel and α7nAChR are involved in the acetylcholine-induced [$Ca^{2+}$]i elevation in the presence of atropine.

Considering these studies elucidating the fundamental properties of nicotine- and acetylcholine-induced [$Ca^{2+}$]i elevation in SHSY-5Y cells, we suggest the following hypotheses (Fig 6). There are two processes of nicotine-induced [$Ca^{2+}$]i elevation: extracellular influx and intracellular mobilization. Some outcomes of the extracellular influx are caused by the nicotine binding-triggered $Ca^{2+}$ influx through nAChRs expressed in the plasma membrane. Other

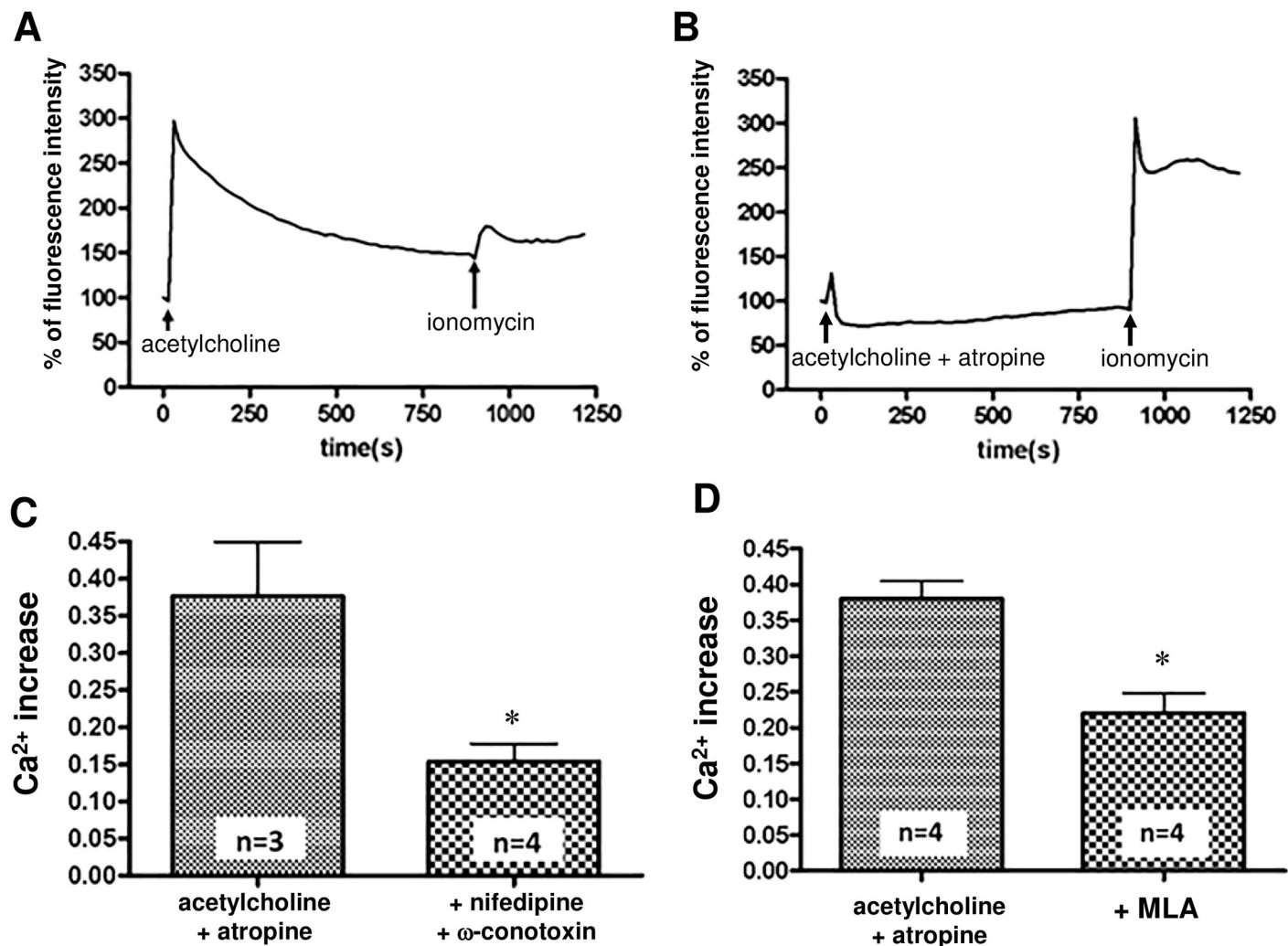

**Fig 5. Properties of acetylcholine-induced [Ca$^{2+}$]i elevation in SH-SY5Y cells.** (A) Temporal changes in [Ca$^{2+}$]i elevation after administration of 10 µM acetylcholine alone. The steep [Ca$^{2+}$]i increase with sustained shape was prominent compared with the nicotine-induced shape. (B) Temporal changes in the [Ca$^{2+}$]i elevation induced by 10 µM acetylcholine in the presence of 1 µM atropine. It is presumed that increases in [Ca$^{2+}$]i are mediated though nAChRs. Compared with the outcome of acetylcholine treatment alone, the [Ca$^{2+}$]i was steeply elevated but was quickly decreased. In both (A) and (B), a representative of 5 independent trials is shown. (C) Involvement of voltage-gated calcium channels in 10 µM acetylcholine and 1 µM atropine-induced [Ca$^{2+}$]i elevation. Pretreatment with 5 µM nifedipine (+ nifedipine) and 1 µM ω-conotoxin GVIA (+ ω-conotoxin) significantly inhibited the acetylcholine and atropine-induced [Ca$^{2+}$]i elevation (D) Involvement of α7nAChR in the [Ca$^{2+}$]i increase induced by 10 µM acetylcholine with 1 µM atropine. Acetylcholine and atropine-induced calcium elevation was significantly suppressed by pretreatment with 5 µM MLA (+MLA). Data indicate the means ± standard error. $^*$ p < 0.05, unpaired t-test, vs the acetylcholine + atropine group.

outcomes are caused by the opening of voltage-gated Ca$^{2+}$ channels, which is triggered by nAChR-induced depolarization. For intracellular mobilization, lipophilic nicotine is thought to cross the cell membrane and mobilize Ca$^{2+}$ from stores, including those in the endoplasmic reticulum, through an undefined mechanism. Currently, it is unknown whether intracellular nAChRs are implicated in Ca$^{2+}$ mobilization.

In contrast to nicotine, nonmembrane-permeable acetylcholine induces [Ca$^{2+}$]i increases via nAChRs and muscarinic receptors expressed in the plasma membrane. As show in Fig 5, our data suggest that voltage-gated Ca$^{2+}$ channel antagonists and an α7nAChR-specific antagonist can be used to distinguish the processes involved in the Ca$^{2+}$ influx mediated through α3

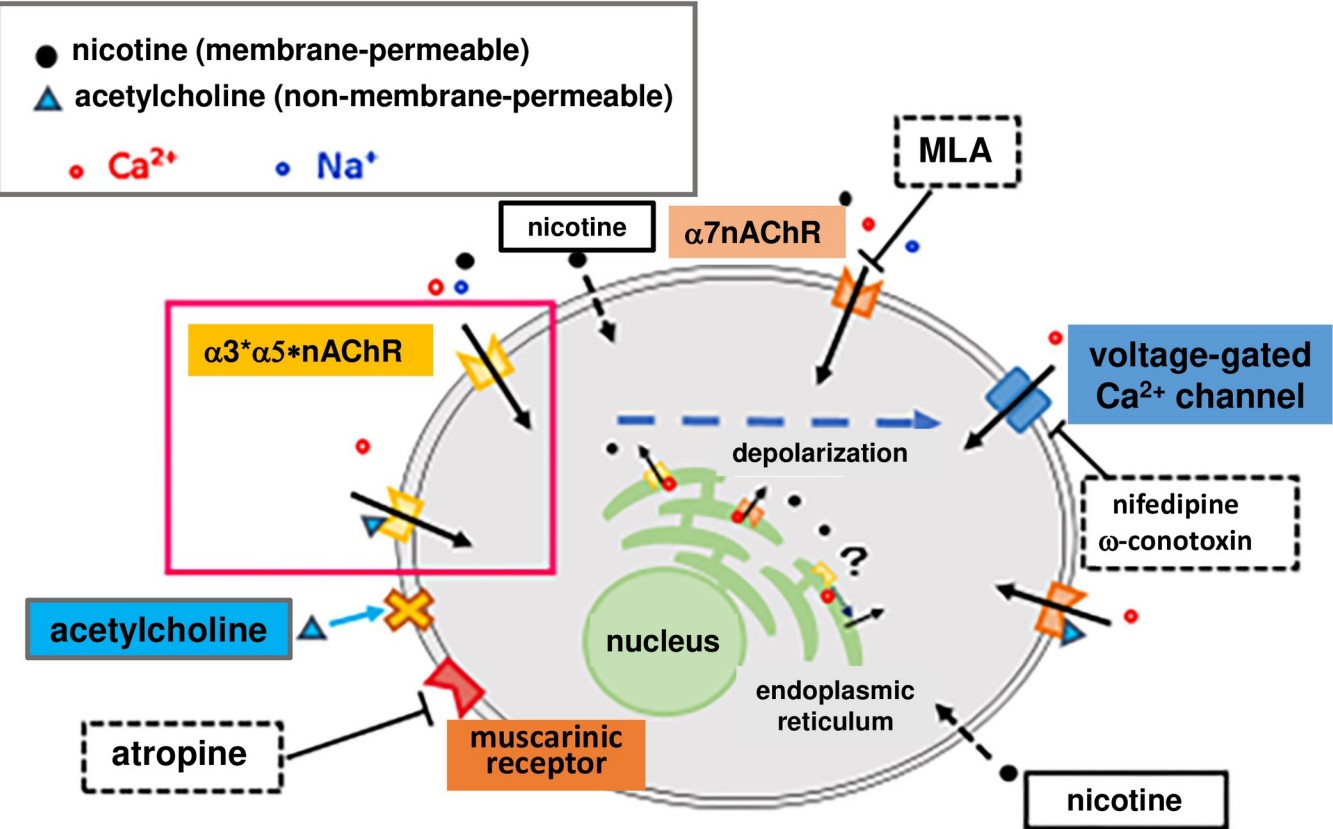

**Fig 6. Fundamental properties of nicotine-induced [Ca$^{2+}$]i elevation in SH-SY5Y cells.** Nicotine may cause an increase in intracellular calcium through three mechanisms: (1) Nicotine binds to α3 $^*$ nAChR, α5 $^*$ nAChR or α7nAChR expressed in the plasma membrane, and calcium and sodium ions enter the cell. (2) The influx of cations causes depolarization, opening the voltage-gated calcium channel, and causes increased Ca$^{2+}$ influx. (3) Lipophilic nicotine penetrates the plasma membrane and enters cells. The internalized nicotine mobilizes Ca$^{2+}$ from stores, including those in the endoplasmic reticulum, by some mechanism. Acetylcholine, with no cell membrane permeability, binds to nicotinic and muscarinic receptors on the plasma membrane, causing an increase in intracellular calcium. We hypothesize that acetylcholine activates only the a3 $^*$ nAChRs and a5 $^*$ nAChRs expressed on the plasma membrane when muscarinic receptors, α7nAChR, and voltage-gated calcium channels are blocked by individual antagonists. In this study, therefore, we observed an increase in intracellular calcium induced by acetylcholine and the inhibitor mix as described in the main text.

$^*$ and α5 $^*$ nAChRs present in the plasma membrane from those mediated by the acetylcholine-induced [Ca$^{2+}$]i elevation in the presence of atropine. Therefore, to elucidate the properties of the α3 $^*$ and α5 $^*$ nAChRs exclusively present in the plasma membrane, we measured acetylcholine (10 μM)-induced [Ca$^{2+}$]i elevation in the presence of atropine (1 μM), MLA (5 μM), nifedipine (5 μM) and ω-conotoxin GVIA (1 μM). We designated the solution containing atropine (1 μM), MLA (5 μM), nifedipine (5 μM) and ω-conotoxin GVIA (1 μM) an inhibitor mix. We consider acetylcholine-induced [Ca$^{2+}$]i elevation in the presence of this inhibitor mix as an α3 $^*$ and α5 $^*$ nAChR-mediated outcome of Ca$^{2+}$ influx.

## Effects of cAMP on [Ca$^{2+}$]i elevation mediated via the α3 $^*$ and α5 $^*$ nAChRs in the plasma membrane

**Effects of dibutyryl cAMP (dbcAMP) treatment on α3 $^*$ and α5 $^*$ nAChR-mediated [Ca$^{2+}$]i elevation.** The effects of cAMP on [Ca$^{2+}$]i elevation mediated via α3 $^*$ and α5 $^*$ nAChRs in the plasma membrane were investigated upon short-term (15 min) or long-term (48 h) treatment of SH-SY5Y cells with dbcAMP (1 mM), a membrane-permeable cAMP

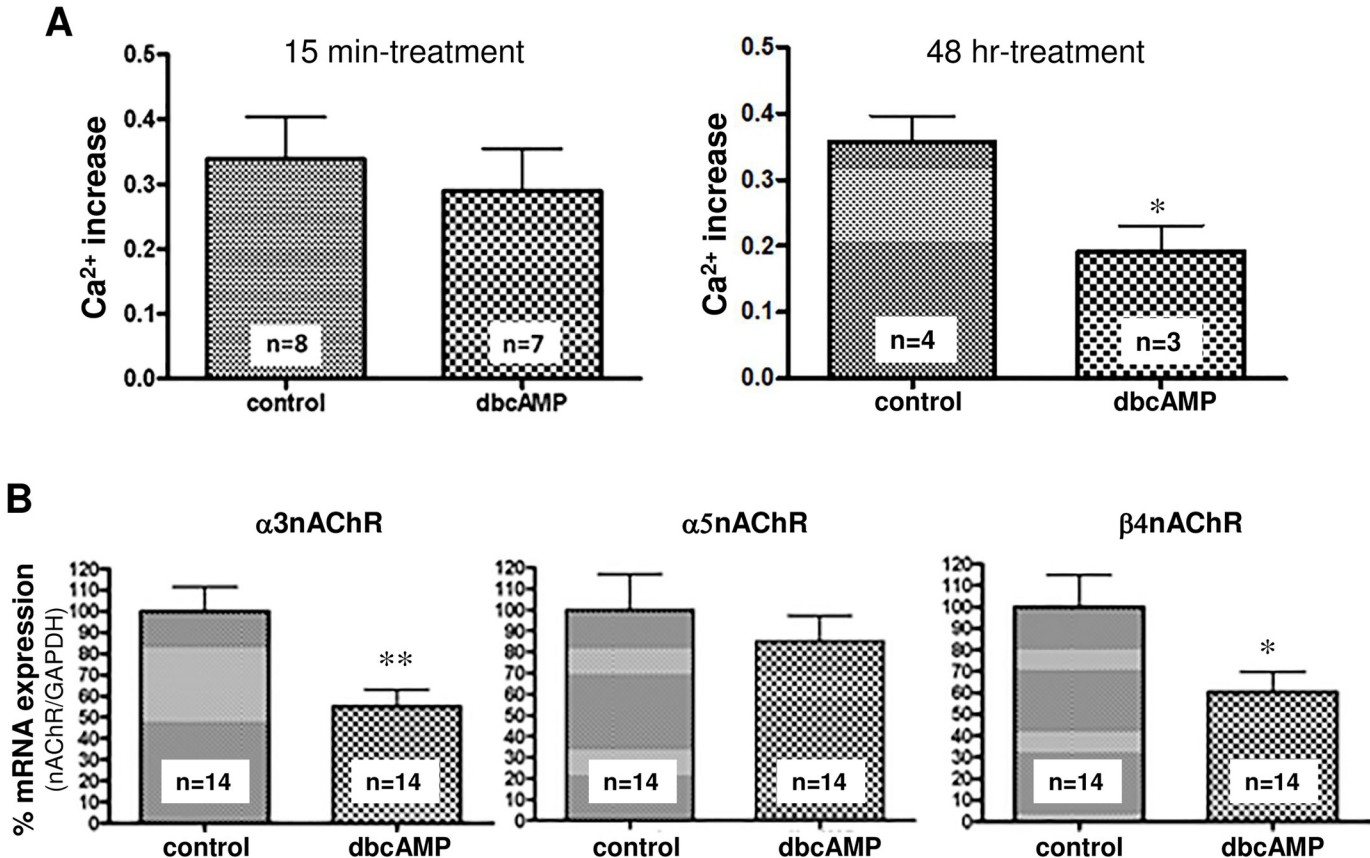

**Fig 7. Effect of dbcAMP treatment on [Ca²⁺]i elevation as induced by acetylcholine and the inhibitor mix.** (A) SH-SY5Y cells were treated with 1 mM dbcAMP for 15 min (left panel) or 48 h (right panel), and the [Ca²⁺]i elevation induced by the acetylcholine and inhibitor mix was examined. Treatment with dbcAMP for 15 min resulted in no changes, but treatment for 48 h significantly inhibited the [Ca²⁺]i elevation. Data represent the means ± standard error. * p < 0.05, unpaired t-test, vs the nontreated control group. (B) Effects of 48 h-treatment with 1 mM dbcAMP on the mRNA expression of various nAChR subunits. Data represent the means ± standard error. * p < 0.05, ** p<0.01, unpaired t-test, vs the nontreated control group.

analog. As shown in Fig 7A, the cells treated with 1 mM dbcAMP for 15 min showed no significant change in the [Ca²⁺]i elevation induced by acetylcholine (10 μM) and the inhibitor mix, whereas 1 mM dbcAMP treatment for 48 h significantly inhibited the increase in[Ca²⁺]i level.

**Effects of dbcAMP treatment on mRNA expression of the nAChR α3, α5 and β4 subunits.** To investigate the mechanism by which the 48-h treatment with 1 mM dbcAMP significantly inhibited the acetylcholine and inhibitor mix-induced [Ca²⁺]i increase, we analyzed the mRNA expression of the α3, α5 and β4 subunits of the nAChRs, which are expressed in SH-SY5Y cells, using real-time PCR. As shown in Fig 7B, the mRNA levels of the α3 and β4 subunits were significantly decreased compared with those in the dbcAMP-naïve control group. These results suggest that long-term dbcAMP treatment suppressed the transcription levels of the α3 and β4 nicotinic receptor subunits, thereby decreasing the influx of Ca²⁺ through the α3 * β4 * nicotinic receptor.

## Discussion

In our study, between 100 and 500 μM nicotine induced a [Ca²⁺]i elevation in SH-SY5Y cells in a concentration-dependent manner (Fig 1). We first examined the origin of calcium ions to

clarify the mechanism by which nicotine evoked $[Ca^{2+}]i$ elevation. For this experiment, we prepared four conditions for extracellular and intracellular calcium (Fig 2). That is, an extracellular buffer in which calcium was replaced with EGTA was used to eliminate extracellular calcium. Thapsigargin was used to deplete calcium in the endoplasmic reticulum, which is the main source of intracellular calcium. After simultaneously completing both of these procedures, we examined the nicotine-induced $[Ca^{2+}]i$ increase in four different intracellular and extracellular calcium conditions (Fig 2). The results suggested that $[Ca^{2+}]i$ elevation had two causes: the influx of calcium from the extracellular space and the mobilization of intracellular calcium, possibly from the endoplasmic reticulum.

To further examine this outcome, we next examined the effects of the nonselective nicotinic receptor antagonist tubocurarine on nicotine-induced $[Ca^{2+}]i$ elevation and nicotine-induced currents. Nicotine is highly lipophilic and readily penetrates the cell membrane. Because membrane impermeable tubocurarine almost completely suppressed nicotine-induced current (Fig 3A), the current influx is suggested be mediated by nAChRs expressed in the plasma membrane. Tubocurarine, however, did not completely suppress the nicotine-induced $[Ca^{2+}]i$ elevation (Fig 3C). This finding suggests that the process insensitive to tubocurarine treatment is the response elicited by intracellularly penetrated nicotine but not the response mediated by nAChRs expressed in the plasma membrane. These findings also support the idea that a component in the intracellular calcium mobilization process contributes to nicotine-induced $[Ca^{2+}]i$ elevation.

As previously reported, the mRNAs for the α3, α5, α7, β2 and β4 subunits of nAChR were expressed in SH-SY5Y cells [22]; therefore, we examined whether these nAChR subunits were involved in the occurrence of nicotine-induced $[Ca^{2+}]i$ elevation using siRNAs of each subunit and specific nAChR inhibitors (Fig 4). The knockdown of the α3 and α5 subunits did not significantly inhibit nicotine-induced $[Ca^{2+}]i$ elevation, whereas the knockdown of the β4 subunit, which is an essential component of α3 * and α5 * nAChRs, significantly inhibited nicotine-induced $[Ca^{2+}]i$ elevation (Fig 4A). The nicotine-induced $[Ca^{2+}]i$ elevation was also significantly inhibited by α-conotoxin AuIB, a specific antagonist of the α3β4 receptor (Fig 4B). In addition, the knockdown of the α7 subunit significantly suppressed the nicotine-induced $[Ca^{2+}]$ elevation (Fig 4C). This strongly suggests the involvement of nAChRs consisting of α3, α5, and α7 subunits in the nicotine-induced $[Ca^{2+}]i$ elevation in SH-SY5Y cells.

Furthermore, since SH-SY5Y cells express voltage-gated L-type and N-type calcium channels [21], the involvement of these channels was investigated using the specific antagonists nifedipine and ω-conotoxin GVIA, respectively. The coadministration of nifedipine and ω-conotoxin GVIA significantly inhibited the increase in nicotine-induced $[Ca^{2+}]i$ level (Fig 4E), suggesting that voltage-gated L-type and N-type calcium channels are also involved in $[Ca^{2+}]i$ elevation.

It is interesting that intracellular calcium mobilization is involved in nicotine-induced $[Ca^{2+}]i$ elevation. It is plausible that intracellularly mobilized $Ca^{2+}$ originates in the endoplasmic reticulum (ER), the main intracellular $Ca^{2+}$ store. It has been reported that α7nAChR expressed in microglia activates PLC and releases $Ca^{2+}$ from IP3 receptors in the endoplasmic reticulum [24]. In addition, calcium-induced calcium release (CICR), which is triggered by $Ca^{2+}$ influx via nAChRs and subsequently activated voltage-gated $Ca^{2+}$ channels, may be involved in nicotine-induced calcium mobilization [20]. These previous studies suggest that $Ca^{2+}$ may also be mobilized via $IP_3$ or ryanodine receptors in the ER by an unidentified mechanism. Alternatively, nicotine-induced $Ca^{2+}$ mobilization was observed even when the nAChR-mediated current was completely blocked by tubocurarine (Fig 3). Additionally, a previous immunohistochemical study revealed the existence of the α3nAChR subunit in the ER of SH-SY5Y cells when α3β4nAChRwas exogenously expressed [25]. These findings suggest that

the penetrated nicotine may directly bind to nAChRs expressed in the ER, recruiting $Ca^{2+}$ to the cytosol through some mechanism.

In contrast to nicotine, acetylcholine, an agonist of the nAChR and muscarinic receptor, does not permeate the cell membrane. We hypothesized that treatment with acetylcholine in the presence of atropine, a muscarinic receptor antagonist, can activate only nAChR, which is expressed in the plasma membrane. We also found that the acetylcholine-induced $[Ca^{2+}]i$ elevation in the presence of atropine also involved the activation of voltage-gated calcium channels and α7nAChR (Fig 5C and 5D).

The aim of this study was to investigate the cAMP-mediated regulation of α3 * and α5 * nAChRs. There are no specific agonists of α3 * and α5 * nAChRs. Thus, it is impossible to observe a nicotine-induced $[Ca^{2+}]i$ increase mediated by α3 * or α5 * nAChR alone in SH-SY5Y cells. Similarly, there are no specific antagonists that act on all types of α3 * and α5 * nAChRs. Thus, we hypothesized that the processes of $[Ca^{2+}]i$ elevation, which are mediated though plasma membrane-expressed α3 * and α5 * nAChRs, can be observed by eliminating processes mediated through the muscarinic receptor, $Ca^{2+}$ channel and α7nAChR that are associated with acetylcholine-induced $[Ca^{2+}]i$ increases (Fig 6); i.e., the $[Ca^{2+}]i$ increases induced by acetylcholine and the inhibitor mix, which contained atropine, nifedipine, ω-conotoxin GVIA and MLA, might involve components associated with α3 * and/or α5 * nAChR-mediated processes.

The $[Ca^{2+}]i$ elevation induced by the acetylcholine and inhibitor mix was attenuated by prolonged treatment (48 h) with dbcAMP but not by treatment with a short duration (15 min) (Fig 7A). This phenomenon might be accompanied by a decrease in the mRNA of the α3 and β4 nAChR subunits (Fig 7B), suggesting that α3 * and α5 * nAChRs are regulated by cAMP at the level of transcription rather than by phosphorylation-related mechanisms.

One limitation of this study is that pharmacological methods have been used to analyze α3 * and α5 * nAChR-mediated reactions. Using siRNAs and specific inhibitors, it seems to be certain that a part of the nicotine response in SH-SY5Y cells is mediated by α3 * and α5 * nAChRs. However, the extent to which the α3 * and α5 * nAChR-mediated processes contribute to the $[Ca^{2+}]i$ elevation induced by acetylcholine and inhibitor mix has not been quantitatively demonstrated. Further studies and methodological breakthroughs are needed. Another limitation is that the concentration of nicotine used in this experiment was relatively high. It is debatable whether the phenomena observed in this study reflect the responses of nicotine under physiological conditions. However, our previous studies have shown that isolated neurons require relatively high concentrations of nicotine to produce nicotine-induced responses [26]. In addition, if nicotine intracellularly accumulates in the brain as a result of chronic smoking or if any native factor with an allosteric effect enhances the response to nicotine, then the data we presented in this study may have physiological or pathological significance.

This study revealed that nicotine causes intracellular calcium elevation by various mechanisms in SH-SY5Y cells. It is plausible that cAMP can regulate nicotine-mediated responses via α3 * and α5 * nAChRs in the habenular nucleus. These results may help to elucidate the role of cAMP-related receptors including GPR3, which are expressed in the medial habenular nucleus, in nicotine dependence.

## Supporting information

**S1 Fig. Effects of 100 μM tubocurarine on the nicotine (250 μM)-induced current.** The representative time plot of leak currents at -70 mV. Leak currents were sampled every ten seconds. Left panel: The bath application of nicotine (250 μM) evoked the large inward current in an SH-SY5Y cell. Nicotine was applied during a period indicated by a black line. Right panel:

Pretreatment with 100 μM tubocurarine (white line) completely suppressed the nicotine-induced current. Representative data from 3experiments are presented.
(TIF)

## Acknowledgments

This work was performed using equipment at the Radiation Research Center for Frontier Science, Natural Science Center for Basic Research and Development, Hiroshima University and Biosignal Research Center, Kobe University.

## Author Contributions

**Conceptualization:** Takayuki Yoshida, Kana Harada, Izumi Hide, Shigeru Tanaka, Norio Sakai.

**Data curation:** Tamayo Takahashi, Takayuki Yoshida, Kana Harada, Tatsuhiko Miyagi, Kouichi Hashimoto.

**Formal analysis:** Kana Harada.

**Funding acquisition:** Izumi Hide, Shigeru Tanaka, Masahiro Irifune, Norio Sakai.

**Investigation:** Tamayo Takahashi, Kana Harada, Norio Sakai.

**Supervision:** Izumi Hide, Shigeru Tanaka, Masahiro Irifune, Norio Sakai.

**Validation:** Norio Sakai.

**Writing – original draft:** Tamayo Takahashi, Takayuki Yoshida.

**Writing – review & editing:** Norio Sakai.

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
