## [Decision Letter · Decision Letter 0]

18 Aug 2020

PONE-D-20-22474

Component of nicotine-induced intracellular calcium elevation mediated though α3- and α5-containing nicotinic acetylcholine receptors are regulated by cyclic AMP　in SH-SY 5Y cells.

PLOS ONE

Dear Dr. Sakai,

Thank you for submitting your manuscript to PLOS ONE. After careful consideration, we feel that it has merit but does not fully meet PLOS ONE’s publication criteria as it currently stands. Therefore, we invite you to submit a revised version of the manuscript that addresses the points raised during the review process.

The manuscript has been assessed by two experts in the field; please find their comments appended at the end of this email. The reviewers have identified several issues, all of which need to be addressed in a revised version of the manuscript. In particular, I would advise you to pay attention to Major comments from Reviewers.

If you will need more time than this to complete your revisions, please reply to this message or contact the journal office at plosone@plos.org. Please include the following items when submitting your revised manuscript:

We look forward to receiving your revised manuscript.

Kind regards,

Nirakar Sahoo, PhD

Academic Editor

PLOS ONE

Journal Requirements:

"This study was supported by a Grant-in-Aid for Scientific Research from the Ministry of Education,

 Sports and Culture and by grants from the Takeda Science Foundation, the Uehara Memorial

Foundation and the Japanese Smoking Research Association. This work was performed using

equipment at the Radiation Research Center for Frontier Science, Natural Science Center for Basic

Research and Development, Hiroshima University and Biosignal 1 Research Center, Kobe University."

"N.S. was supported by  JSPS KAKENHI Grant Numbers 19H03409, 16H05133, 16K15318 and 25293061.

I. H. was supported by   JSPS KAKENHI Grant Numbers 18K06891  and 15K08234.

S. T. was supported by JSPS KAKENHI Grant Numbers 18K07392 and     15K09317."

Reviewers' comments:

Reviewer's Responses to Questions

**Comments to the Author**

1. Is the manuscript technically sound, and do the data support the conclusions?

Reviewer #1: Yes

Reviewer #2: Yes

2. Has the statistical analysis been performed appropriately and rigorously? 

Reviewer #1: Yes

Reviewer #2: Yes

3. Have the authors made all data underlying the findings in their manuscript fully available?

Reviewer #1: Yes

Reviewer #2: Yes

4. Is the manuscript presented in an intelligible fashion and written in standard English?

Reviewer #1: Yes

Reviewer #2: Yes

5. Review Comments to the Author

Reviewer #1: In this article entitled “Component of nicotine-induced intracellular calcium elevation mediated though �3- and �5-containing nicotinic acetylcholine receptors are regulated by cyclic AMP in SH-SY 5Y cells”, Takahashi and colleagues have evaluated the mechanistic aspects of regulation of intracellular Ca2+ ([Ca2+]i) by nicotine acetylcholine receptors (nAChRs). The authors tested nicotine and acetylcholine dependent elevation of [Ca2+]i levels as well as effect of cAMP using SHSY-5Y cell line. The authors observed nicotine and acetylcholine induced increase in [Ca2+]i levels, that nicotine dependent Ca2+ elevation was a result of both extracellular and intracellular mobilization of Ca2+ ions and that dibutyryl cAMP (dbcAMP), a cAMP analog, could suppress transcription of α3 and β4 subunits of nAChR.

In general, the results shown agree with the conclusions drawn and data has been presented in a clear and easily understandable manner. However, the authors need to address the issues described below:

1) In section 2-3, effect of tubocurarine was tested on [Ca2+]i levels of cells treated with 250 µM nicotine. A complete inhibition of inward currents was observed while [Ca2+]i was significantly reduced but not complete. Could the authors comment on why 250 µM nicotine was used rather than 500 µM. This is important since (a) in Section 2-2 (Fig. 2), to check the source of [Ca2+]i the authors used 500 µM nicotine and (b) effect of 500 µM nicotine on [Ca2+]i (Fig. 1B) was much more pronounced compared to 250 µM. Verifying the effect of 500 µM nicotine is therefore important, especially on the current.

2) Based on Fig. 1A, expression of α7 in SHSY-5Y cells is significantly lower than α3 and α5, which raises the question whether this cell line is an ideal system for studying the involvement of α7nAChR in regulating [Ca2+]i levels.

Grammatical errors / typos:

1) Typo in the title of the article “though” should be replaced by “through”.

2) Page 12, Line 4 – Fig. 4E has been mistakenly typed as Fig. 2E

Reviewer #2: Takahashi et al. investigate the role of nicotine and specific nicotinic acetylcholine receptor (nAChRs) subunits on extracellular and intracellular calcium mobilization. Nicotine is involved in signaling in the central nervous system and is known to mediate addictive behavior. Thus, it is important to study the underlying effects of nicotine in the CNS using a cell culture model (SH-SY5Y).

The authors used various molecules that release intracellular calcium stores, including agonists and antagonists of nAChRs, muscarinic AChRs and voltage-gated calcium channels (VGCCs). The authors ultimately identified that the alpha-7 subunit of nAChR was important in mediated calcium mobilization. The authors proposed that nicotine affects cellular calcium response at the plasma membrane level through calcium influx through VGCCs. They also suggested that nicotine, being membrane-permeable, can induce intracellular calcium release through an unknown mechanism based on their results.

Overall, the strength of this study is the thoughtful and logical approaches they have implemented to answer their questions. Appropriate controls were included in all experiments. The authors also addressed the limitations of the study and discussed their results appropriately.

Issues that need to be addressed to improve the manuscript:

Page 3, Line 23: The authors should define “Gs” before using the abbreviation here or elsewhere.

Page 5, Line 24: For real-time qPCR using the Livak method (2^-deltadelta Ct), the authors should follow the MIQE (Minimum Information required for publication of Q-PCR Experiments) guidelines (Bustin et al. (2009), Clin Chem, 55:611-622) by providing the primer efficiencies and R-squared of the PCR.

Page 7, Line 13: The authors should indicate how many independent trials were done, and how many cells were voltage-clamped to arrive with the data they presented on Fig. 3A.

Pages 24-28, Figure Legends: The authors should consistently indicate or mention the independent trials or cell numbers if applicable. For example, on figure 1, the legend does not provide experimental trials for the left panel of Fig. 1B, but they show on the right panel n=6 for 100 uM, n=4 for 250 uM, n=4 for 500 uM). It is important to mention the trials every time especially that they mentioned that the data are represented as means +/- SEM.

Also, the same issue apply in other figures where the authors showed %fluorescence or trace, but they did not mention on the Fig Legend the experimental trials done.

6. PLOS authors have the option to publish the peer review history of their article (what does this mean?). If published, this will include your full peer review and any attached files.

Reviewer #1: No

Reviewer #2: No

---

## [Author Response · Author response to Decision Letter 0]

29 Sep 2020

Overall revision

We have added two authors, Takayuki Yoshida and Kana Harada, who did the experiments for revising manuscript.

Reply to reviewer #1

We appreciate the comments that lead to the improvement of this manuscript.

1) In section 2-3, effect of tubocurarine was tested on [Ca2+]i levels of cells treated with 250 µM nicotine. A complete inhibition of inward currents was observed while [Ca2+]i was significantly reduced but not complete. Could the authors comment on why 250 µM nicotine was used rather than 500 µM. This is important since (a) in Section 2-2 (Fig. 2), to check the source of [Ca2+]i the authors used 500 µM nicotine and (b) effect of 500 µM nicotine on [Ca2+]i (Fig. 1B) was much more pronounced compared to 250 µM. Verifying the effect of 500 µM nicotine is therefore important, especially on the current.

According to the reviewer’s comment, we investigated the effects of tubocurarine on the currents induced by 500 µM nicotine. As shown in revised figure 3A, 100 µM tubocurarine almost completely suppressed the current induced by 500 µM nicotine as well as the currents induced by 250 µM nicotine as shown in a former figure 3A. In addition, quantitative analysis of tubocurarine effects was added as Figure 3B. 

A former figure 3A showing the effect of tubocurarine on the 250 µM nicotine-induced current has been presented as a supplemental figure 1.

We have observed nicotine-induced calcium elevation at concentrations up to 500 µM. Generally, pharmacological studies using inhibitors and siRNAs are usually performed at intermediate concentrations (preferably at ED50 concentrations) rather than at the highest concentrations tested. Therefore, in this study, we examine the effects of various inhibitors and siRNAs on nicotine-induced calcium elevation at the concentration of 250 µM. However, in the experiments shown in Figure 2, we had to observe a significant nicotine-induced calcium elevation under various conditions, so we used 500 µM nicotine, at the highest concentration of nicotine tested.

The description in the main text has been changed in accordance with the change in Figure 3 as follows.

Material methods section; page 7, line 16- page 8, line 1, page 8, line6-7 

Results section; page 10, line18-24

Figure legends; page 25, line 11-20 

2) Based on Fig. 1A, expression of α7 in SHSY-5Y cells is significantly lower than α3 and α5, which raises the question whether this cell line is an ideal system for studying the involvement of α7nAChR in regulating [Ca2+]i levels.

In this study, the mRNA expression of each nAChR in SHSY5Y cells was examined only once. Therefore, three new cases were examined and the results was presented as a new Figure 1A. Again, α7 expression appears to be lower than α3 and α5 expressions in the graph of new figure 1A. However, the efficiency of the primers to detect each nAChR subunit and the efficiency of the cDNA synthesis of each nAChR subunit in the process of synthesizing cDNA from mRNA are not same between each subunit. Therefore, we cannot simply judge from this graph that the expression of α7 is lower than those of α3 and α5.

A PubMed search yielded 88 papers investigating nicotine-induced calcium elevation via the α7 nAChR in SHSY-5Y cells. This fact does not suggest that SHSY-5Y cells are not ideal for studying nicotine-induced calcium elevation via the α7 nAChR.

Grammatical errors / typos:

1) Typo in the title of the article “though” should be replaced by “through”.

2) Page 12, Line 4 – Fig. 4E has been mistakenly typed as Fig. 2E

We corrected these errors in the revised manuscript.

Reply to reviewer #2

We appreciate the comments that lead to the improvement of this manuscript.

-Page 3, Line 23: The authors should define “Gs” before using the abbreviation here or elsewhere.

We have changed the description “Gs” to “Gs (stimulatory G protein)”. (page 3, line 23-24)

- Page 5, Line 24: For real-time qPCR using the Livak method (2^-deltadelta Ct), the authors should follow the MIQE (Minimum Information required for publication of Q-PCR Experiments) guidelines (Bustin et al. (2009), Clin Chem, 55:611-622) by providing the primer efficiencies and R-squared of the PCR.

We noticed that the delta Ct method was used in this study, rather than the 2^-deltadelta Ct method, to assess mRNA expression, so we have changed the description to reflect that. Information for the primer efficiencies and R-squared of the PCR have been also described in revised manuscript. (page 6, line 6-9)

And, we re-performed the RT-PCR studies three times because the experiment was done only once in a former manuscript. The new results have been presented as figure 1A.

- Page 7, Line 13: The authors should indicate how many independent trials were done, and how many cells were voltage-clamped to arrive with the data they presented on Fig. 3A.

We newly investigated the effect of tubocurarine on the current induced by 500 µM nicotine. We performed 8 independent experiments. The descriptions were changed accordingly. 

Material methods section; page 7, line 16- page 8, line 1, page 8, line6-7 

Results section; page 10, line18-24

Figure legends; page 25, line 11-20 

- Pages 24-28, Figure Legends: The authors should consistently indicate or mention the independent trials or cell numbers if applicable. For example, on figure 1, the legend does not provide experimental trials for the left panel of Fig. 1B, but they show on the right panel n=6 for 100 µM, n=4 for 250 µM, n=4 for 500 µM). It is important to mention the trials every time especially that they mentioned that the data are represented as means +/- SEM.

In accordance with the reviewer's suggestion, we have added a description concerning the number of experiments and the representation of the data. (page 24, line 5-6, line 11, page 25, line 4-5, page 25, line 15, page 25, line 24) 

With regard to data representation, we have already described them as “Data represent the means ± standard error”.

-Also, the same issue applies in other figures where the authors showed %fluorescence or trace, but they did not mention on the Fig Legend the experimental trials done.

Concerning the trace using %fluorescence, we have already described the information in materials methods and figure legends section as bellow. 

The time-dependent change in intracellular calcium ion [Ca2+]i was determined by setting the fluorescence intensity to 100% before administration of nicotine or acetylcholine. (page 6, line23 -page 7, line 1)

The fluorescence before nicotine administration was assumed to be 100%, and the change in [Ca2+]i after application of nicotine at each concentration is shown ( page 24, line 9-10)

---

## [Decision Letter · Decision Letter 1]

2 Nov 2020

Component of nicotine-induced intracellular calcium elevation mediated through a3- and a5-containing nicotinic acetylcholine receptors are regulated by cyclic AMP　in SH-SY 5Y cells.

PONE-D-20-22474R1

Dear Dr. Sakai,

We’re pleased to inform you that your manuscript has been judged scientifically suitable for publication and will be formally accepted for publication once it meets all outstanding technical requirements.

Kind regards,

Nirakar Sahoo, PhD

Academic Editor

PLOS ONE

Additional Editor Comments (optional):

Reviewers' comments:

Reviewer's Responses to Questions

**Comments to the Author**

1. If the authors have adequately addressed your comments raised in a previous round of review and you feel that this manuscript is now acceptable for publication, you may indicate that here to bypass the “Comments to the Author” section, enter your conflict of interest statement in the “Confidential to Editor” section, and submit your "Accept" recommendation.

Reviewer #1: All comments have been addressed

Reviewer #3: All comments have been addressed

2. Is the manuscript technically sound, and do the data support the conclusions?

Reviewer #1: Yes

Reviewer #3: Yes

3. Has the statistical analysis been performed appropriately and rigorously? 

Reviewer #1: Yes

Reviewer #3: Yes

4. Have the authors made all data underlying the findings in their manuscript fully available?

Reviewer #1: Yes

Reviewer #3: Yes

5. Is the manuscript presented in an intelligible fashion and written in standard English?

Reviewer #1: Yes

Reviewer #3: Yes

6. Review Comments to the Author

Reviewer #1: (No Response)

Reviewer #3: There are no further concern.

Authors have improved their manuscript according to previous comments by reviewers.

7. PLOS authors have the option to publish the peer review history of their article (what does this mean?). If published, this will include your full peer review and any attached files.

Reviewer #1: No

Reviewer #3: **Yes: **Abdul Qadir Syed

---

## [Editor Report · Acceptance letter]

16 Nov 2020

PONE-D-20-22474R1 

Component of nicotine-induced intracellular calcium elevation mediated through α3- and α5-containing nicotinic acetylcholine receptors are regulated by cyclic AMP　in SH-SY 5Y cells. 

Dear Dr. Sakai:

I'm pleased to inform you that your manuscript has been deemed suitable for publication in PLOS ONE. Congratulations! Your manuscript is now with our production department. 

Kind regards, 

on behalf of

Dr. Nirakar Sahoo 

Academic Editor

PLOS ONE